# Modeling of Personalized Treatments in Colon Cancer Based on Preclinical Genomic and Drug Sensitivity Data

**DOI:** 10.3390/cancers13236018

**Published:** 2021-11-30

**Authors:** Marlen Keil, Theresia Conrad, Michael Becker, Ulrich Keilholz, Marie-Laure Yaspo, Hans Lehrach, Moritz Schütte, Johannes Haybaeck, Jens Hoffmann

**Affiliations:** 1Experimental Pharmacology and Oncology Berlin-Buch GmbH (EPO), Robert-Roessle-Str. 10, 13125 Berlin, Germany; m.keil@epo-berlin.com (M.K.); theresia.conrad@epo-berlin.com (T.C.); mibecker@epo-berlin.com (M.B.); 2Comprehensive Cancer Center, Charite-Universitätsmedizin, Chariteplatz 1, 10117 Berlin, Germany; ulrich.keilholz@charite.de; 3Department of Computational Molecular Biology and Department of Vertebrate Genomics/Otto Warburg Laboratory Gene Regulation and Systems Biology of Cancer, Max Planck Institute for Molecular Genetics, Ihnestrasse 73, 14195 Berlin, Germany; yaspo@molgen.mpg.de (M.-L.Y.); lehrach@molgen.mpg.de (H.L.); 4Alacris Theranostics GmbH, Max-Planck-Straße 3, 12489 Berlin, Germany; m.schuette@alacris.de; 5Institute of Pathology, Neuropathology, and Molecular Pathology, Medical University of Innsbruck, 6020 Innsbruck, Austria; johannes.haybaeck@i-med.ac.at; 6Diagnostic & Research Center for Molecular Biomedicine, Institute of Pathology, Medical University of Graz, 8036 Graz, Austria

**Keywords:** colon cancer, personalized treatment, drug combinations

## Abstract

**Simple Summary:**

This experimental preclinical study developed a strategy to identify signatures for the personalized treatment of colon cancer focusing on target-specific drug combinations. Tumor growth inhibition was analyzed in a preclinical phase II study using 25 patient-derived xenograft models (PDX) treated with drug combinations blocking alternatively activated oncogenic pathways. Results reveal an improved response by combinatorial treatment in some defined molecular subgroups and potential alternative treatment options in KRAS- and BRAF-mutated colon cancer.

**Abstract:**

The current standard therapies for advanced, recurrent or metastatic colon cancer are the 5-fluorouracil and oxaliplatin or irinotecan schedules (FOxFI) +/− targeted drugs cetuximab or bevacizumab. Treatment with the FOxFI cytotoxic chemotherapy regimens causes significant toxicity and might induce secondary cancers. The overall low efficacy of the targeted drugs seen in colon cancer patients still is hindering the substitution of the chemotherapy. The ONCOTRACK project developed a strategy to identify predictive biomarkers based on a systems biology approach, using omics technologies to identify signatures for personalized treatment based on single drug response data. Here, we describe a follow-up project focusing on target-specific drug combinations. Background for this experimental preclinical study was that, by analyzing the tumor growth inhibition in the PDX models by cetuximab treatment, a broad heterogenic response from complete regression to tumor growth stimulation was observed. To provide confirmation of the hypothesis that drug combinations blocking alternatively activated oncogenic pathways may improve therapy outcomes, 25 models out of the well-characterized ONCOTRACK PDX panel were subjected to treatment with a drug combination scheme using four approved, targeted cancer drugs.

## 1. Introduction

Although KRAS and BRAF mutations have been established as biomarkers for cetuximab resistance in colorectal cancer (CRC), the predictive value is not satisfying. Approximately 35% of the KRAS wild-type (wt) population does not respond to cetuximab, while there is growing evidence that some mutant tumors might respond to the treatment. As cetuximab is usually combined with FOxFI, it is difficult to define the contribution to the overall response [1]. However, in some colon cancer PDX models treated with single cetuximab, almost complete regressions were observed [2], raising questions as to whether the combination with FOxFI is mandatory for all patients.

Within the ONCOTRACK project [2], drug sensitivity to the EGFR antibody cetuximab was determined in a cohort of 58 colon cancer PDX models, derived from 58 patients with primary or metastatic cancer. In parallel, these PDX models were treated with the recently approved VEGF and multikinase inhibitor regorafenib and two further investigational drugs targeting the mammalian target of rapamycin (mTOR) and mitogen-activated protein kinase (MAP) pathways, an experimental mTOR inhibitor (BI mTOR FR), and the experimental MEK inhibitor AZD6244. However, no drug combination effects were evaluated in this project.

To identify a rationale for drug combinations, we selected a subpanel of 25 PDX models from the ONCOTRACK cohort with a heterogeneous genetic profile as well as sensitivity towards the four drugs for further analysis.

When comparing the response to the four drugs, the following questions were raised: There is a population of PDX where a strong response to cetuximab is observed—would this group of tumors still require combination therapy, and is there an additional molecular predictor for this subgroup other than the KRAS or BRAF wt phenotype?A second group of PDX seems to significantly benefit from one of the treatments, but still slowly progressing in growth—the most obvious question for this subgroup is will they profit from a combination?Lastly, there is the group of treatment-resistant tumors mainly with mutant KRAS or BRAF and usually worse prognoses—are there any new hypotheses/rationales for a combination of treatments?

To address these questions, a pilot drug combination study was initiated using 25 PDX models representing each of the three response groups. As, during the ONCOTRACK project, two experimental drugs were used, which are not available for clinical routine, we decided to perform the combination study with the approved drugs cetuximab (targeting EGFR), trametinib (targeting MEK), regorafenib (targeting multiple kinases, i.e., VEGFR1/2/3, TIE2, KIT, RET, BRAF, BRAFV600E and FGFR-1, PDGFR-ß), and everolimus (targeting mTOR) (Figure 1). The use of clinically approved drugs should allow the better translation of the experimental results to the clinical setting.

## 2. Materials and Methods

### 2.1. Patient Samples

Patient samples were obtained as described by Schütte et al. [2]. Samples were obtained and stored according to the current good clinical practice (GCP) guidelines. Informed consent was obtained from all human subjects included in the study. The study was approved by the local Institutional Review Board of Charité University Medicine (Charité Ethics Cie: Charitéplatz 1, 10117 Berlin, Germany (EA 1/069/11)) and the ethics committee of the Medical University of Graz (Ethic commission of the Medical University of Graz, Auenbruggerplatz 2, 8036 Graz, Austria), confirmed by the ethics committee of the St. John of God Hospital Graz (23-015 ex 10/11).

### 2.2. Development and Characterization of Patient-Derived Xenografts (PDX)

Development of the PDX models was described in detail by Schütte et al. [2]. In brief, resected tumor tissues were transplanted to immunodeficient mice (NMRI nude or NOG, Taconic, Bomholdtgard, DK-Tac: NMRI-Foxn1nu, females, 6–8 weeks at start of transplantation). Animal experiments were carried out in accordance with the United Kingdom Coordinating Committee on Cancer Research regulations for the Welfare of Animals and of the German Animal Protection Law and approved by the local responsible authorities. Mice were monitored 3 times weekly for tumor engraftment for up to 3 months. Engrafted tumors at a size of approximately 1 cm^3^ were surgically excised and smaller fragments re-transplanted to naive NMRI nu/nu mice for further passage. Within passage 1 to 3, numerous samples were cryo-conserved (dimethylsulfoxide medium) for further experiments. Tumors were passaged no more than 6 times. For confirmation of tumor histology, tumor tissue was formalin-fixed and paraffin-embedded (FFPE) and 5 µm sections were prepared. Samples were stained according to a standard protocol for hematoxilin, eosin, and Ki67 to ensure xenograft comparability to the original specimen. Cases with changed histological pattern were sent for pathological review and outgrowth of lymphoproliferative disorders was excluded.

### 2.3. Molecular Characterization

Molecular characterization was performed within the ONCOTRACK project as described by Schütte et al. [2] and data were used in this study. In brief, DNA and RNA obtained from the PDX tumor sample were analyzed for gene expression, copy number variants, somatic SNVs, gene fusions [2]. Microsatellite status was analyzed using the five monomorphic markers BAT25, BAT26, NR21, NR24, and NR27 and pentaplex polymerase chain reactions (PCR) [2].

### 2.4. In Vivo Drug Response Testing of the Xenografts

Twenty-five xenografts of the 58 PDX models from the ONCOTRACK cohort were included in the drug combination studies. Response to the selected compounds and combinations was evaluated in early passages using the design of a preclinical phase II study. Tumor fragments of similar size were transplanted subcutaneously to a cohort of mice. At palpable tumor size (80–150 mm^3^), mice were randomized to treatment or control groups consisting of 3 animals each. Doses and schedules were chosen according to previous experience in animal experiments and represent maximum tolerated or efficient doses. The following drugs, doses, and schedules for single and combination treatments were used: Cetuximab (Merck KGA), 30 mg/kg biweekly intraperitoneally, in saline;Regorafenib (Bayer AG), 10 mg/kg once daily orally, in pluronic F68 and PEG400;Everolimus (Novartis), 3 mg/kg once daily orally, in Tween 80 and saline;Trametinib (Selleckchem), 3 mg/kg once daily orally, in hydroxypropylmethylcellulose and Tween 80 in water for injection.

Drugs were obtained from the pharmacy or Selleckchem, Houston, TX, USA. The injection volume was 0.1 mL/20 g body weight. In case of therapy resistance (no regression or stable disease), selected mice received all four drugs until further progression for another 4 weeks. Treatment was continued till tumor size exceeded 1.5 cm^3^ or animals showed loss of >15% body weight. From the first treatment day onwards, the tumor volumes and body weights were recorded twice weekly. At the end of the treatment period, animals were sacrificed, and blood and tumor samples were collected and stored in liquid nitrogen immediately. 

Animal welfare was controlled twice daily. Tumor volume (TV) was calculated from the length and width of subcutaneous tumors (TV = (length × [width]^2^)/2). Sensitivity to the tested compounds was determined as tumor growth inhibition by treatment in comparison to the control (T/C) at each measurement point. Efficacy of the tested drugs in PDX models was classified using the adopted clinical response criteria (RECIST). We calculated the relative tumor volume (RTV) as the ratio of the TV on the last day before the study ended or start of quadruple treatment/TV on the first day of treatment.

The response criteria, taking as reference the baseline sum diameters, were defined as follows:Strong Response: (RTV < 1.6);Moderate Response: (RTV < 2.5);Minor Response: (RTV < 5.5);Resistance: (RTV > 5.5).

As RTV is a condensed summary parameter, no standard deviation values for replicate measurements are given in the Appendix A; these values have been determined and are available in the raw data. 

### 2.5. Statistical Analyses of Mutational Load and Drug Sensitivity Values

Statistical and graphical analyses were performed with Prism version 9.1.0 (GraphPad Software, San Diego, CA, USA). Statistical differences were analyzed by unpaired t tests with Welch’s correction (comparison of the mutational load in MSI and MSS PDX models), Kruskal–Wallis test with Dunn’s multiple comparisons post-test, and one-way ANOVA with Holm–Šídák’s multiple comparisons post-test (comparison of treatment-dependent RTV values in the respective subgroups). Regarding the comparison of treatment-dependent RTV values in subgroup I, PDX Co11672-327 was excluded from the analysis, since it was the sample with an activating BRAF mutation. *p* values are displayed as follows: *p* value > 0.05 ns; *p* value ≤ 0.05 *; *p* value ≤ 0.01 **; *p* value ≤ 0.001 ***.

## 3. Results

The selected panel of 25 colon cancer PDX models with heterogeneous sensitivity towards cetuximab was tested for response to the single drugs and the drug combinations with cetuximab and trametinib, cetuximab and regorafenib, and cetuximab with everolimus. These combinations should provide a parallel blockade of targets in the downstream MAP kinase pathway or the PI3K/AKT pathway (Figure 1). Mice were treated for up to 4 weeks to determine the initial response to the mono- and dual combination therapies. Responses (RTVs) to the single drug and drug combination treatments are summarized in Figure 2. In the case of therapy resistance (no regression or stable disease), selected mice received all four drugs until further progression for another 4 weeks. General health status and body weights were recorded on a regular basis daily or twice weekly as toxicity of drug combinations has been reported frequently. In our studies, the drugs were well-tolerated in the groups treated with cetuximab, regorafenib, and trametinib. Everolimus has caused a reversible minor body weight loss of between 5 and 10% in some studies. The drug combination treatments were tolerated without additional toxicity (representative data are shown in Appendix A). 

Based on the analysis of a panel of 65 genes most frequently mutated in the selected 25 PDX models, four molecular subgroups were determined (Figure 2 and Figure 3). The Consensus Molecular Subtype (CMS) classification describes four CRC subtypes with distinct biological characteristics that show prognostic and potential predictive value in the clinical setting. As already described by Schuette et al. [2] for the ONCOTRACK panel of colon cancer PDX models, the CMS classification cannot be transferred directly to the colon cancer PDX models as the immune cell components are missed by xenotransplantation in immune-suppressed mice. Nevertheless, our groups shared some similar characteristics to the annotation within the CRC consensus molecular group labels (CMS1 to CMS4) [3]. 

The first group (*n* = 5) is characterized by microsatellite instability (MSI) status, accompanied by a statistically significant higher mutational load in the selected panel of 65 genes (>10 mutations in the panel) compared to PDX models with microsatellite stability (MSS) status (Figure 2). However, in contrast to CMS1, we found a BRAF mutation in only 2 out of these 5 PDX models. In the second cohort (*n* = 5), all colon cancers had a BRAF V600E mutation; however, again, in contrast to CMS1, all tumors were MSS and had otherwise a very low frequency of mutations (≤3 mutations in the panel). The third group (*n* = 7) was KRAS and BRAF wt, characterized by adenomatous polyposis coli (APC) and p53 mutations and a low frequency of other mutations (Figure 2). This group seemed to correspond with the CMS2 (canonical) subgroup characteristics. The last group (*n* = 8) had KRAS mutations and some frequent co-mutations (Figure 2). This group was MSS and included characteristics from both the CMS3 and CMS4 subgroups (Figure 3). 

### 3.1. Patterns of Drug Response in Colon Cancer PDX Models in Relation to the Four Molecular Subgroups

To understand potential synergies by the combination treatments in correlation to the molecular subgroups, drug response evaluation was performed for each described subgroup. 

#### 3.1.1. Subgroup I—MSI Hypermutated

Subgroup I with MSI and high mutation frequency is mainly resistant to the four targeted drugs, with only some minor responses. Although there is a trend for synergistic effects by the combination of cetuximab and trametinib, these differences are not statistically relevant (Figure 4a). Taken together, this molecular subgroup is rather resistant to both single and combination treatment with targeted drugs (Appendix A).

The PDX model Co11672-327 seems to be an exceptional model in this subgroup due to the uncommon kinase impaired BRAF G446V mutation. Hence, it was excluded from the above-mentioned analyses of treatment responses (Figure 4a). However, this model is very sensitive toward all three combinations. It has been reported that other PDX with this BRAF mutation have been sensitive to EGFR and MEK inhibition with even stronger activity of combinations [4].

#### 3.1.2. Subgroup II—MSS BRAF-Mutated

Subgroup II is characterized by the BRAF V600E mutation. However, all five PDX are MSS and have a low frequency of mutations, a rather uncommon combination when compared to the classification in CMS1. Interestingly, all models are resistant to cetuximab, but rather sensitive to the MEK inhibitor trametinib, with a significant difference in terms of responses (Figure 4b). The effect of trametinib in 4 out of 5 PDX models—Co10629-150, Co10786-181, Co10979-212, Co11388-289—is long-lasting disease stabilization with partial regression for up to 3 months (Appendix A). Regorafenib and everolimus as single drugs have some minor activity, which is, however, only for regarofenib statistically significant different from cetuximab. The combination of cetuximab with both drugs has synergistic effects and is significantly better. Combination of trametinib with cetuximab does not improve response in the four strongly responding models; however, it has synergistic effects in Co11336-283, the model with only a minor response to trametinib (Appendix A).

#### 3.1.3. Subgroup III—MSS KRAS and BRAF Wild-Type

The third group is MSS and does not have prominent oncogenic mutations in the MAP kinase pathway (KRAS and BRAF). All seven colon cancers share common molecular characteristics, with a generally low number of mutations, mainly APC and p53 and three models with PIK3CA/B mutations. All seven colon cancers are highly sensitive to cetuximab, with a strong response. Surprisingly, all models are rather resistant to trametinib and everolimus. The combination with the three other drugs in general did not further improve the response (Figure 4c). However, in selected models, combination with trametinib (Co10849-191, Co11192-259) exerted synergistic effects and complete tumor regressions were observed (Appendix A). The combination of cetuximab with everolimus (mTOR inhibition) showed a tendency for synergistic effects in the same two colon cancer PDX models, where, in particular, two complete regressions were remarkable.

The colon cancer model Co11291-273 is characterized by a RET mutation. While both drugs cetuximab or regorafenib alone can induce a strong response, the combination, however, shows synergistic effects with partial to complete regression of the tumors (Appendix A). 

#### 3.1.4. Subgroup IV—MSS KRAS-Mutated

The fourth subgroup consists of eight colon cancer models with KRAS mutations, and all of them are MSS. Besides the KRAS mutations, these eight colon cancer PDX models do share other common molecular characteristics: some have mutations in PI3K (4), in RET (2), and SMAD4 (1). In general, the number of mutations is higher when compared to Subgroup II.

All models are resistant to treatment with cetuximab and to the inhibition of the MAPK pathway by the MEK inhibitor trametinib. However, a statistically significant synergistic activity of the combination of cetuximab with trametinib was reported in this subgroup (Figure 4). The combination achieved four stable disease states—Co11319-280, Co10748-171, Co10501-118, Co10803-183—and two minor responses—Co11300-277 and Co10412-128—in these otherwise highly resistant tumors (Appendix A). 

Whereas everolimus is not active in any of these models, response to regorafenib is heterogeneous, with three colon cancers showing a response Co11319-280, Co10501-118, Co11300-277, and Co10412-128, whereas the other four are resistant. Combination with cetuximab seems to provide some significant synergistic effects in selected models. The PDX model Co11993-352, for example, is rather resistant to all single treatments and combinations, except for cetuximab and regorafenib (Appendix A).

## 4. Discussion

Whereas, in the “pre-” personalized medicine area, doublet chemotherapy was seen as the standard of care for advanced or metastatic colon cancer, the identification of activated signaling via the MAPK pathway or the VEGF receptor family in colon cancer has introduced new treatment opportunities with monoclonal antibodies or kinase inhibitors [5]. As per current clinical guidelines, pan-RAS, BRAF, HER2, and mismatch repair (MMR) status are established molecular markers for selection of patient treatment [6]. However, it has been realized that the predictive value of these biomarkers has limitations. From the group of patients without mutations in the MAP kinase pathway members KRAS and BRAF, significant sub-fractions do not benefit from EGFR antibody treatment.

Patients with KRAS or BRAF mutations continue to have a very poor prognosis, often with median survival of less than 12 months, and treatment options are still limited [5]. Approximately 7–10% of CRC patients have a mutation of the BRAF gene, with 90% of them displaying the V600E [7,8,9]. There are three classes of BRAF mutations; while class I and II, i.e., with the most common mutation—V600E—have a notably worse prognosis, class III has impaired kinase activity and better prognosis. Thus, for patients with BRAF- or KRAS-mutated colon cancer, alternative targeted treatment strategies still need to be developed [10]. Before the era of targeted therapy combinations, intense chemotherapy with anti-VEGF was the standard of care in patients with BRAF class I and II mutations [11]. While KRAS has been, until recently, seen to be undruggable and the new KRAS inhibitors target only the less frequent mutation G12C [12], several BRAF and MEK kinase inhibitors have been developed and tested also in colon cancer patients. The effects of BRAF inhibitors such as vemurafenib in melanoma treatment raised some expectations for the treatment of colon cancer patients. However, inhibition of the MAPK pathway with single BRAF inhibitors such as vemurafenib or dabrafenib has not demonstrated therapeutic benefits in clinical trials [13,14]. It has become clear that BRAF inhibition in colon cancer can lead to activation of EGFR through an ERK-dependent negative feedback loop and induce further upregulation of other receptor tyrosine kinases, including the other human epidermal growth factor receptors, or activation of the phosphoinositide 3-kinase (PI3K)/AKT/mTOR pathway [15,16]. Activation of the PI3K/AKT/mTOR pathway has also been implicated in BRAF inhibition resistance [17]. 

Based on these findings, we have evaluated combination therapies blocking both pathways with four different drugs in our representative preclinical models to generate new hypotheses for treatments that will overcome resistance and improve response in selected colon cancer patient subgroups (Figure 1). As colon cancers express high levels of activated EGFR, a combined blockade of EGFR and BRAF or even downstream MEK may work synergistically and could be a potential therapeutic opportunity in CRC. As the combination of cetuximab with vemurafenib has not been very effective, we chose either the approved MEK inhibitor trametinib or the (pan) BRAF and VEGF kinase inhibitor regorafenib for the combination experiments. Regorafenib inhibits, next to the main targets, several other kinases, such as TIE2, KIT, RET, RAF-1, BRAFV600E, PDGFR, and FGFR.

As mentioned earlier, activation of the PI3K/AKT/mTOR pathway by negative feedback is one potential pathway in cetuximab resistance. We therefore included the mTOR inhibitor everolimus as the third combination partner in the studies.

The BEACON trial [18] first demonstrated that both a dual therapy targeting BRAF (encorafenib) and EGFR (cetuximab) and especially a triple combination targeting BRAF (encorafenib), MEK (binimetinib), and EGFR (cetuximab) can increase the survival of colon cancer patients compared to the current standard of care (SoC). We observed that tumors in Subgroup II with BAF V600E as a potential single driver mutation strongly responded to trametinib. The combination of trametinib with cetuximab seemed to further increase the overall response in this colon cancer subgroup. Our findings are in line with the results from the BEACON study; however, they provide additional findings that might be of clinical relevance for the treatment of these patients.

For selection of the optimal treatment of colon cancer patients with BRAF mutations, the MSI status might be considered as an additional biomarker. BRAF mutations have been observed in 30–50% of MSI-high CRC, compared with 10% in microsatellite stable tumors [19,20]. According to our data, tumors with BRAF V600E mutations and MSS status (low mutational rate) seem to strongly benefit from the combination of trametinib (MEK inhibition) and cetuximab (EGFR inhibition), whereas MSI tumors seem not to have a benefit, although the number of models in this cohort is too low for statistically significant conclusions.

Colon cancers without KRAS and BRAF mutations strongly respond to treatment with cetuximab and are rather resistant to the other three treatments. However, findings of mutations in other oncogenic pathways have been reported in this subgroup. For example, RET is altered in 2.94% of colorectal carcinoma patients [21] and, further, PI3K signaling can be activated by direct mutation or amplification of PIK3CA or loss of PTEN. Approximately 40% of CRC have been shown to have alterations in PI3K pathway genes, which are almost always mutually exclusive from each other [22].

We observed, in selected models, synergistic effects and complete tumor regressions by the combination of cetuximab with everolimus—for example, in model Co11192-259 with a mutation in PIK3CB. These findings lead to the hypothesis that the combination of cetuximab with PI3K or mTOR inhibition might be of benefit for a subgroup with wt KRAS and BRAF, but activation of the PI3K pathway.

The combination of cetuximab with regorafenib (RAF, VEGF, and RET inhibition) did show synergistic effects in model Co11291-273 with RET mutation. This combination might be a potential therapeutic option for the subgroup of patients with RET as a driver mutation, as we observed complete regressions under this treatment combination (Appendix A).

All PDX models with KRAS mutations tested in our study were resistant to treatment with cetuximab, confirming, in this case, the good predictivity of KRAS mutations as a biomarker for cetuximab in colon cancer patients. Similarly, the inhibition of the MAPK pathway by the MEK inhibitor trametinib was not effective. The most surprising outcome in this subgroup was the statistically significant synergistic activity of the combination of cetuximab with trametinib. The combination achieved four stable disease cases and two minor responses in these otherwise highly resistant tumors (Appendix A). As there are currently no other treatment alternatives for this colon cancer subgroup, in cases with MSS status, further evaluation of this combination inhibiting EGFR and MEK might be considered.

The MSI-high subgroup with or without BRAF mutation is resistant to the tested targeted drugs. Based on the MSI and the hypermutated profile, this group would be better treated with chemo- or immune therapies. Data from the ONCOTRACK project confirm this hypothesis at least in part, as strong sensitivity to the SoC chemotherapies 5-FU or irinotecan has been observed in 4 out these 5 models [2]. A combination of chemotherapy with a PARP inhibitor has recently shown activity in preclinical models of peritoneal metastases of colorectal cancer with a similar molecular profile [23] and might be an alternative opportunity for this subgroup of patients. Currently, immunotherapy is evaluated as a therapeutic option in these subtypes [24]. As immunotherapies cannot be evaluated in the common PDX models on immunodeficient mice, further studies in humanized mouse models might be required [25].

## 5. Conclusions

Molecular profiling allows the identification of colon cancer subgroups for personalized treatment.PDX models of CRC enable preclinical screening of targeted drugs and identification of synergistic combinations in correlation with molecular profiles.Microsatellite stable colon cancer models with BRAF or KRAS mutations in Subgroups II and IV have shown responses to the combination of EGFR (cetuximab), MEK (trametinib), and/or RAF (regorafenib) inhibition, providing a strong hypothesis for further evaluation.PI3K, mTOR (everolimus), and RET (regorafenib) inhibition seem to be synergistic with EGFR (cetuximab) inhibition in selected colon cancers with those activated pathways.Although the small preclinical phase II-like sample size precludes firm conclusions, the results of the study have revealed interesting potential relations, which either might be followed up in a larger preclinical panel or translated in personalized clinical trials.

## Figures and Tables

**Figure 1 cancers-13-06018-f001:**
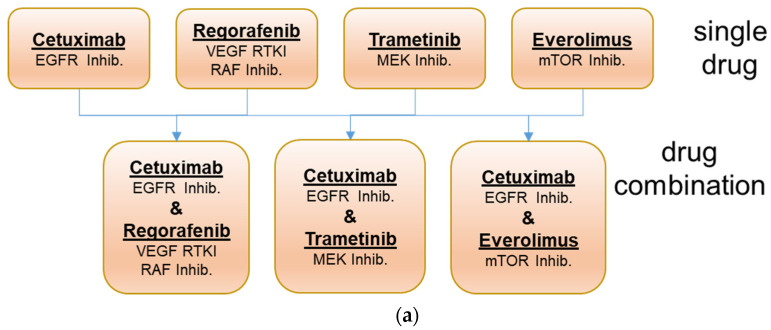
Drug combinations (**a**) and pathways blocked by the different targeted drugs (**b**).

**Figure 2 cancers-13-06018-f002:**
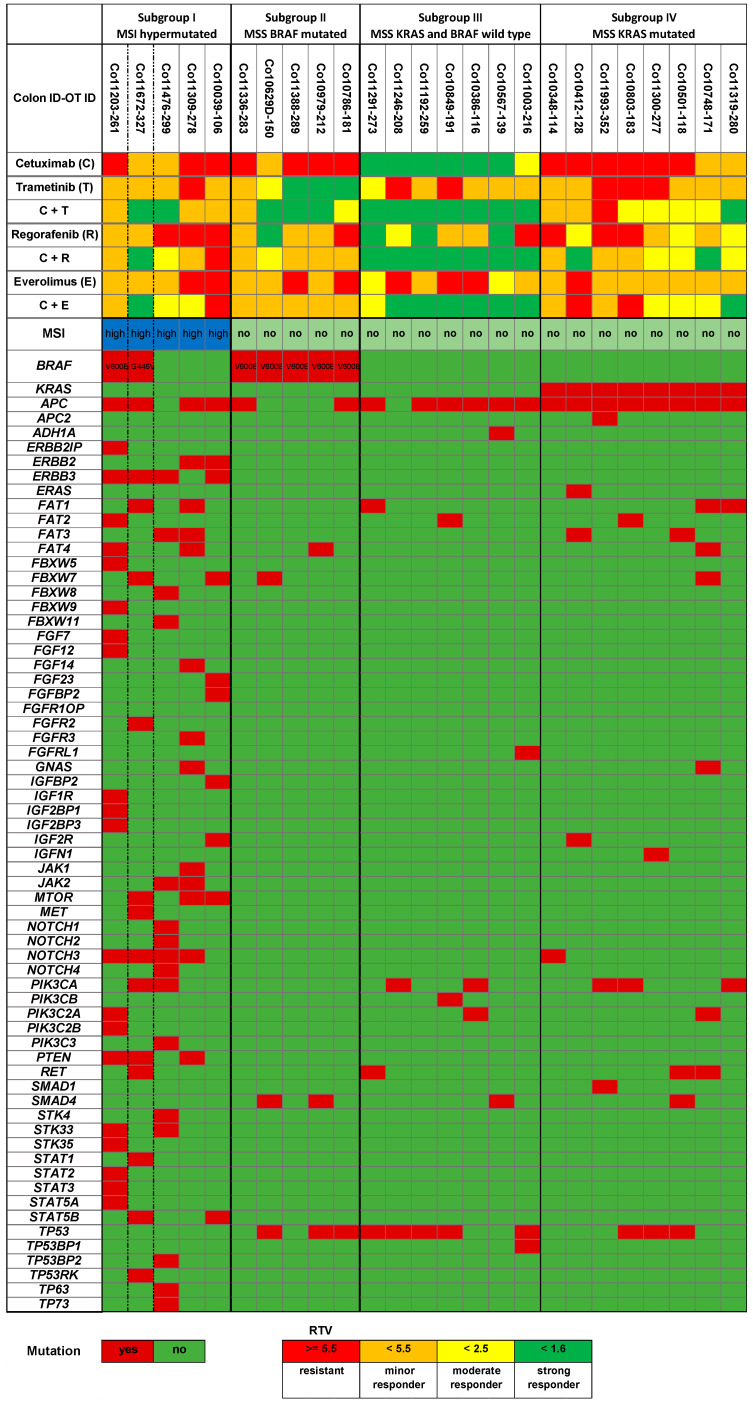
Response of 25 colon cancer PDX to cetuximab, regorafenib, trametinib, and everolimus in correlation with genetic mutation profile. Response data are provided as RTV values (RTV = quote of TV on the last day before study ended or start of quadruple treatment/TV on the first day of treatment).

**Figure 3 cancers-13-06018-f003:**
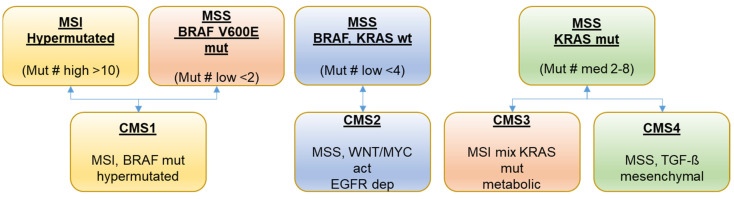
Genomic classification of colon cancer subgroups.

**Figure 4 cancers-13-06018-f004:**
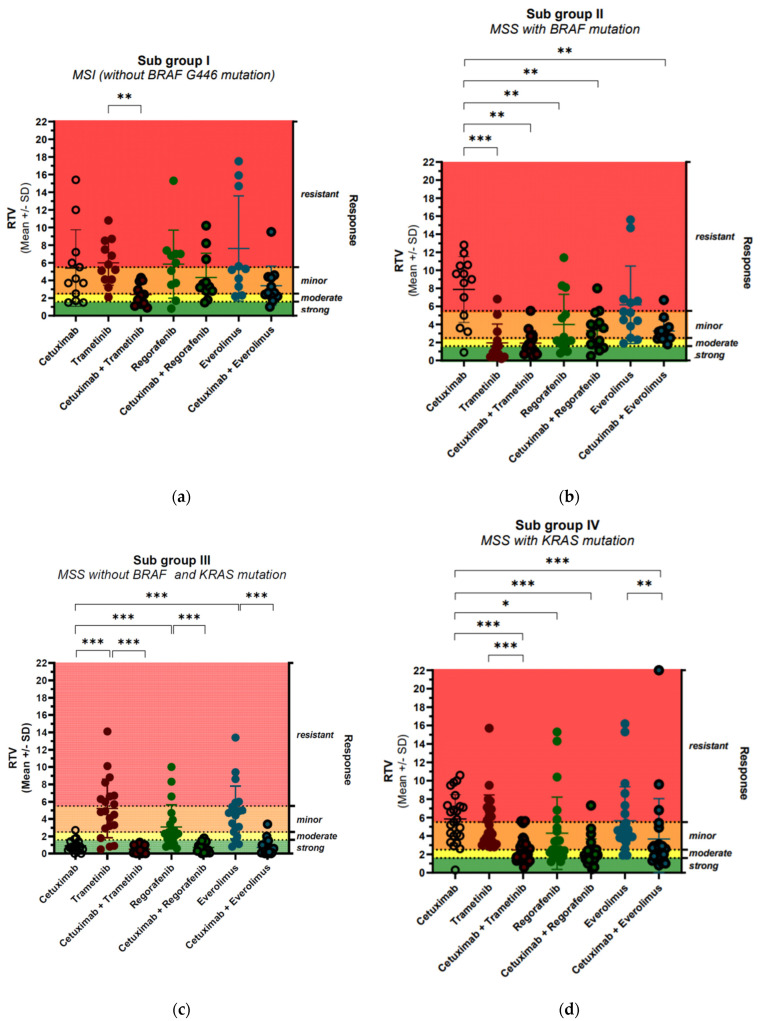
Effects of single treatments in comparison to drug combinations in: Subgroup I with MSI hypermutated colon cancer (**a**), Subgroup II with MSS and BRAF-mutated colon cancer (**b**), Subgroup III with MSS and KRAS and BRAF wild-type colon cancer (**c**), Subgroup IV with MSS and KRAS-mutated colon cancer (**d**) (*p* values are displayed as follows: *p* value > 0.05 ns; *p* value ≤ 0.05 *; *p* value ≤ 0.01 **; *p* value ≤ 0.001 ***).

## Data Availability

The complete set of NGS data for patient tumors has been published by Schütte et al. [2] and is available in the European Genome-Phenome Archive (EGA) of the EBI data repository under accession number EGAS00001001752.

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
