# Peer review of "Modeling of Personalized Treatments in Colon Cancer Based on Preclinical Genomic and Drug Sensitivity Data"

_cancers, 2021, doi:10.3390/cancers13236018_

Round 1
Reviewer 1 Report
Minor grammatical corrections:
line 20- Correct ‘growth inhibition in was’ to ‘growth inhibition was’.
line 43- Please add ‘Wild-type’ before using the acronym.
line 47- Correct ‘complete regressions are observed’ to ‘complete regressions were observed’.
line 48- Correct ‘mandatory for all populations’ to ‘mandatory for all patients’.
line 48- Correct ‘genetic profile and as well as’ to ‘genetic profile as well as’.
line 61- Correct ‘require still combination therapy’ to ‘still require combination therapy’.
line 62- Correct ‘next to KRAS or BRAF’ to ‘apart from/other than KRAS or BRAF’.
line 66- Correct ‘Lately there’ to ‘Lastly there’.
line 69- Correct ‘study has been initiated’ to ‘study was initiated’.
line 75- Consider replacing the sentence ‘This should allow a better translation of the experimental results in clinical hypotheses’ to ‘Use of clinically approved drugs should allow a better translation of the experimental results in clinical settings’.
line 89- Figure-1 Correct ‘two drug combination’ to ‘three drug combination’.
line 89- Figure-1 Replace ‘Mek inhib’ to ‘MEK inhib’.
line 94- Consider adding ‘Schütte M et al. [2]’ instead of just referencing [2].
lines 145, 146- Correct ‘We calculated the relative tumor volume (RTV) ratio of the TV of the last day before study end or start of quadruple treatment / TV of the first day of treatment’ to ‘We calculated the relative tumor volume (RTV) as the ratio of the TV on the last day before study ended or start of quadruple treatment / TV on the first day of treatment’.
line 181- Consider changing ‘our groups are sharing’ to ‘our groups shared’.
line 283- Consider changing ‘do not benefit of EGFR’ to ‘do not benefit from EGFR’.
line 311- Correct ‘we chooses either’ to ‘we chose either’.
line 321- Please describe the ‘SoC’ before using the acronym.
line 335- Please use one format for KRAS. In several places, it is written as kRAS while in others it is written as KRAS.
Content comments:
- Figure 2 is not described anywhere in the text. Also, the way PDX groups are organized in the table is opposite to how they are described in the text. For example, the authors have described the MSI group first while that group is represented in the end (left-most) part of the table. It might also help to annotate groupings on the top of the table (Subgroup 1, Subgroup 2, Subgroup 3, Subgroup 4) to make it easier for the readers to grasp the information.
- The authors have mentioned ‘synergism’ ‘additive’ in the manuscript but have not described how did they evaluated for these drug combination effects.
- One of the major concerns in combination therapies is toxicity. The author shave used not only ‘2 drug’ combination but also ‘4 drug combination’ (in cases of resistance). However, the authors have not commented how did the mice tolerate the 4-drug combination? The authors should show the mice weight and other panels to show the effects of drug combination on mice health.
- Was there any improvement in the mice with 4-drug combination?
Author Response
Dear Sir or Madam,
Thank you very much for your kind review of our manuscript. Your comments greatly helped to clarify and to improve our research article leading to a substantial revision including the incorporation of additional data.
Please kindly find our answers to your comments in detail below.
Comments 1: Minor grammatical corrections:
line 20- Correct ‘growth inhibition in was’ to ‘growth inhibition was’.
line 43- Please add ‘Wild-type’ before using the acronym.
line 47- Correct ‘complete regressions are observed’ to ‘complete regressions were observed’.
line 48- Correct ‘mandatory for all populations’ to ‘mandatory for all patients’.
line 48- Correct ‘genetic profile and as well as’ to ‘genetic profile as well as’.
line 61- Correct ‘require still combination therapy’ to ‘still require combination therapy’.
line 62- Correct ‘next to KRAS or BRAF’ to ‘apart from/other than KRAS or BRAF’.
line 66- Correct ‘Lately there’ to ‘Lastly there’.
line 69- Correct ‘study has been initiated’ to ‘study was initiated’.
line 75- Consider replacing the sentence ‘This should allow a better translation of the experimental results in clinical hypotheses’ to ‘Use of clinically approved drugs should allow a better translation of the experimental results in clinical settings’.
line 89- Figure-1 Correct ‘two drug combination’ to ‘three drug combination’.
line 89- Figure-1 Replace ‘Mek inhib’ to ‘MEK inhib’.
line 94- Consider adding ‘Schütte M et al. [2]’ instead of just referencing [2].
lines 145, 146- Correct ‘We calculated the relative tumor volume (RTV) ratio of the TV of the last day before study end or start of quadruple treatment / TV of the first day of treatment’ to ‘We calculated the relative tumor volume (RTV) as the ratio of the TV on the last day before study ended or start of quadruple treatment / TV on the first day of treatment’.
line 181- Consider changing ‘our groups are sharing’ to ‘our groups shared’.
line 283- Consider changing ‘do not benefit of EGFR’ to ‘do not benefit from EGFR’.
line 311- Correct ‘we chooses either’ to ‘we chose either’.
line 321- Please describe the ‘SoC’ before using the acronym.
line 335- Please use one format for KRAS. In several places, it is written as kRAS while in others it is written as KRAS.
Answer to comments 1:
All suggested grammatical corrections have been implemented in the manuscript, in Figure 1 “two drug combinations” has been replaced by “drug combinations”
Content comment 2:
- Figure 2 is not described anywhere in the text. Also, the way PDX groups are organized in the table is opposite to how they are described in the text. For example, the authors have described the MSI group first while that group is represented in the end (left-most) part of the table. It might also help to annotate groupings on the top of the table (Subgroup 1, Subgroup 2, Subgroup 3, Subgroup 4) to make it easier for the readers to grasp the information.
Answer to comment 2
Figure 2 is now described in the text, PDX groups have been organized as suggested with annotation on the top.
Content comment 3:
- The authors have mentioned ‘synergism’ ‘additive’ in the manuscript but have not described how did they evaluated for these drug combination effects.
Answer to comment 3
We have deleted the term “additive” and only discuss synergism. A synergistic effect is defined for the significant difference comparing the single therapies and the combination.
Content comment 4:
- One of the major concerns in combination therapies is toxicity. The author shave used not only ‘2 drug’ combination but also ‘4 drug combination’ (in cases of resistance). However, the authors have not commented how did the mice tolerate the 4-drug combination? The authors should show the mice weight and other panels to show the effects of drug combination on mice health.
Answer to comment 4
We have included comments on the tolerability of the combination treatments and included some representative graphs on the body weights in the suppl. Figures.
Content comment 5:
- Was there any improvement in the mice with 4-drug combination?
Answer to comment 5
The 4-drug combination has not been evaluated systematically, rather selected groups have been treated with the 4 drug combination. The data are given in the suppl Figures. In some cases we see tumor growth regression after the initiation of the 4-drug treatment. However, as we do not have an additional control to continuing the 2 drug combination, one can not statistically analyze the data. The 4 drug combinations have rather been performed to generate preliminary information for further combination experiments.

Reviewer 2 Report
This manuscript entitled “Modeling of personalized treatments in colon cancer based on 2 preclinical genomic and drug sensitivity data” by Marlen Keil and Jens Hoffman et al have tried to figure out the significance of the combination therapy rather than the single drug in colon cancer to four groups of patient derived xenograft (PDX) consisting of MSI hypermutation, MSS BRAF mutation, MSS KRAS and BRAF wild type, and MSS KRAS mutation respectively after molecular profiling. As a combination therapy, the authors have generated three groups of combination by adding cetuximab (C) with other three targeted drugs namely regorafenib (R), trametinib (T), and everolimus (E) independently and compared their effectiveness against the single drug itself. It has been found that the MSS KRAS and BRAF mutations strongly response to the combination of C+T indicating that this combination might play a pivotal role in colon cancer patients having the MSS KRAS and BRAF mutation after further study. Surely this work is helpful for the physicians and other researchers working in this arena to select the combination therapy for colon cancer patients. Before publishing this great piece of work, I request some minor changes to the authors.
Minor revisions:
- I would like to suggest the authors to elaborate on all the abbreviations earlier either in the Introduction section or can generate another section “naming elaboration of the abbreviation” before the introduction section so that it will be more understandable to the readers.
- In section 2, could you please generate another section namely ‘Materials” include all the reagents used in this study with their specification and sources?
- In the section 2.4, the authors have mentioned 0.1 ml/20g body weight. I suggest to present the exact dose of all individual drugs itself and in the combination.
- Please include the supplementary Figure 3 (for the subgroup III) in the main manuscript. Also mention and number the supplementary figures as well.
- In the conclusion section, I strongly suggest the authors to mention in the beginning which combination therapy e.g., C+T, C+R or C+ E has more effectiveness to the subgroup III or IV. Here the authors have concluded by mentioning the inhibition mechanism of the drugs which is not easily understandable to al readers.
Author Response
Dear Sir or Madam,
Thank you very much for your kind review of our manuscript. Your comments greatly helped to clarify and to improve our research article leading to a substantial revision including the incorporation of additional data.
Please kindly find our answers to your comments in detail below.
This manuscript entitled “Modeling of personalized treatments in colon cancer based on 2 preclinical genomic and drug sensitivity data” by Marlen Keil and Jens Hoffman et al have tried to figure out the significance of the combination therapy rather than the single drug in colon cancer to four groups of patient derived xenograft (PDX) consisting of MSI hypermutation, MSS BRAF mutation, MSS KRAS and BRAF wild type, and MSS KRAS mutation respectively after molecular profiling. As a combination therapy, the authors have generated three groups of combination by adding cetuximab (C) with other three targeted drugs namely regorafenib (R), trametinib (T), and everolimus (E) independently and compared their effectiveness against the single drug itself. It has been found that the MSS KRAS and BRAF mutations strongly response to the combination of C+T indicating that this combination might play a pivotal role in colon cancer patients having the MSS KRAS and BRAF mutation after further study. Surely this work is helpful for the physicians and other researchers working in this arena to select the combination therapy for colon cancer patients. Before publishing this great piece of work, I request some minor changes to the authors.
Comment 1: Minor revisions:
- I would like to suggest the authors to elaborate on all the abbreviations earlier either in the Introduction section or can generate another section “naming elaboration of the abbreviation” before the introduction section so that it will be more understandable to the readers.
Answer to comment 1:
We have checked all abbreviations and included naming in the manuscript, i.e. for SoC, MAP, mTOR. FOxFI.
Comment 2: Minor revisions:
- In section 2, could you please generate another section namely ‘Materials” include all the reagents used in this study with their specification and sources?
Answer to comment 2:
We have included the following information about the used reagents in the manuscript:
The following drugs, doses and schedules for single and combination treatments were used:
Cetuximab (Merck KGA), 30 mg/kg biweekly intraperitoneally, in saline
Regorafenib (Bayer AG), 10 mg/kg once daily orally, in pluronic F68 & PEG400
Everolimus (Novartis), 3 mg/kg once daily orally, in Tween 80 & saline
Trametinib (Selleckchem), 3 mg/kg once daily orally, in hydroxypropylmethyl-cellulose & Tween 80 in water for injection
Drugs were obtained from the pharmacy or Selleckchem, Houston, TX, USA The injection volume was 0.1 ml/20 g body weight.
Comment 3: Minor revisions:
- In the section 2.4, the authors have mentioned 0.1 ml/20g body weight. I suggest to present the exact dose of all individual drugs itself and in the combination.
Answer to comment 3:
We have included the information about the applied doses in the manuscript:
The following drugs, doses and schedules for single and combination treatments were used:
Cetuximab (Merck KGA), 30 mg/kg biweekly intraperitoneally, in saline
Regorafenib (Bayer AG), 10 mg/kg once daily orally, in pluronic F68 & PEG400
Everolimus (Novartis), 3 mg/kg once daily orally, in Tween 80 & saline
Trametinib (Selleckchem), 3 mg/kg once daily orally, in hydroxypropylmethyl-cellulose & Tween 80 in water for injection
Drugs were obtained from the pharmacy or Selleckchem, Houston, TX, USA The injection volume was 0.1 ml/20 g body weight.
Comment 4: Minor revisions:
- Please include the supplementary Figure 3 (for the subgroup III) in the main manuscript. Also mention and number the supplementary figures as well.
Answer to comment 4:
The supplementary figures have been revised, we have in addition included representative information about the tolerability of the combination by showing some representative body weight graphs. The supplementary figures are now with titles and numbers. We like the idea to implement supplementary figure 3 in the main manuscript, we can do this easily. I would like to leave this decision to the editors.
Comment 5: Minor revisions:
- In the conclusion section, I strongly suggest the authors to mention in the beginning which combination therapy e.g., C+T, C+R or C+ E has more effectiveness to the subgroup III or IV. Here the authors have concluded by mentioning the inhibition mechanism of the drugs which is not easily understandable to al readers.
Answer to comment 5:
We have included the subgroups and the drug names as follows:
- Microsatellite stable colon cancer models with BRAF or KRAS mutations in subgroups II and IV have shown responses to the combination of EGFR (cetuximab), MEK (trametinib), and/or RAF (regorafenib) inhibition, providing a strong hypothesis for further evaluation
- PI3K, mTOR (everolimus) and RET (regorafenib) inhibition seems to be additive with EGFR (cetuximab) inhibition in selected colon cancers with those activated pathways

Reviewer 3 Report
The authors use a small PDX panel to examine pairs of drugs for efficacy across 5 signature types of colon cancer. The results demonstrate the potential power of the methodology, but the small sample size precludes firm conclusions (suggestive). However, I find the paper interesting as a demonstration piece. In the conclusion, I think the authors might discuss their future plans in using this information and better explain the limitation of this example of the methodology.
Author Response
Dear Sir or Madam,
Thank you very much for your kind review of our manuscript. Your comments greatly helped to clarify and to improve our research article leading to a substantial revision including the incorporation of additional data.
Please kindly find our answers to your comments in detail below.
Comment 1:
The authors use a small PDX panel to examine pairs of drugs for efficacy across 5 signature types of colon cancer. The results demonstrate the potential power of the methodology, but the small sample size precludes firm conclusions (suggestive). However, I find the paper interesting as a demonstration piece. In the conclusion, I think the authors might discuss their future plans in using this information and better explain the limitation of this example of the methodology.
Answer to comment 1:
The authors are aware of the limitations of the study due to the small sample size. In accordance with the reviewer we see rather the potential of the methodology to provide preclinical data with high evidence to generate hypotheses for further translational clinical research.
We have included the following in the conclusion:
- Although the small preclinical phase II like samples size precludes firm conclusions, results from the study have revealed interesting potential relations which either might be followed up in a larger preclinical panel or translated in personalized clinical trials.
